Dynamic multi-species occupancy models reveal individualistic habitat preferences in a high-altitude grassland bird community

Maphisa David H. Maphisad@gmail.com 1 2 3
Smit-Robinson Hanneline 4 5
Altwegg Res 2 6
1 Statistical Ecology program, South African National Biodiversity Institute , Cape Town , South Africa
2 Department of Statistical Sciences, Statistics in Ecology, Environment and Conservation, University of Cape Town , Cape Town , South Africa
3 Ingula Partnership Project, Blairgowrie, Randburg, BirdLife South Africa , Johannesburg , South Africa
4 BirdLife South Africa , Johannesburg , South Africa
5 Applied Behavioural Ecological & Ecosystem Research Unit (ABEERU), UNISA , Johannesburg , South Africa
6 African Climate and Development Initiative, University of Cape Town , Rondebosch, Cape Town , South Africa
Boyer Alison
Electronic publication date: 2019 Feb 15
Publication date: 2019
Volume: 7
Electronic Location ID: e6276
Received 2018 May 9; Accepted 2018 Dec 12
Copyright: ©2019 Maphisa et al.
Copyright year: 2019
Copyright holder: Maphisa et al.
License: This is an open access article distributed under the terms of the Creative Commons Attribution License, which permits unrestricted use, distribution, reproduction and adaptation in any medium and for any purpose provided that it is properly attributed. For attribution, the original author(s), title, publication source (PeerJ) and either DOI or URL of the article must be cited.
License URL: https://creativecommons.org/licenses/by/4.0/

Keywords: Hierarchical occupancy models, Habitat suitability, Grazing and fire, Grass height and cover, Monitoring, Occupancy, Habitat management

Funding: Eskom through The Ingula Partnership Mazda Wildlife Fund National Research Foundation of South Africa The first author was supported in the position of BirdLife South Africa Ingula Project Manager with funding by Eskom through The Ingula Partnership. The Mazda Wildlife Fund supported the first author with a vehicle for the duration of the project, while employed by BirdLife South Africa. Res Altwegg was supported by the National Research Foundation of South Africa. The funders had no role in study design, data collection and analysis, decision to publish, or preparation of the manuscript.

==============================
Moist, high-altitude grasslands of eastern South African harbour rich avian diversity and endemism. This area is also threatened by increasingly intensive agriculture and land conversion for energy production. This conflict is particularly evident at Ingula, an Important Bird and Biodiversity Area located within the least conserved high-altitude grasslands and which is also the site of a new Pumped Storage Scheme. The new management seeks to maximise biodiversity through manipulation of the key habitat variables: grass height and grass cover through burning and grazing to make habitat suitable for birds. However, different species have individual habitat preferences, which further vary through the season. We used a dynamic multi-species occupancy model to examine the seasonal occupancy dynamics of 12 common grassland bird species and their habitat preferences. We estimated monthly occupancy, colonisation and persistence in relation to grass height and grass cover throughout the summer breeding season of 2011/12. For majority of these species, at the beginning of the season occupancy increased with increasing grass height and decreased with increasing grass cover. Persistence and colonisation decreased with increasing grass height and cover. However, the 12 species varied considerably in their responses to grass height and cover. Our results suggest that management should aim to provide plots which vary in grass height and cover to maximise bird diversity. We also conclude that the decreasing occupancy with increasing grass cover and low colonisation with increasing grass height and cover is a results of little grazing on our study site. We further conclude that some of the 12 selected species are good indicators of habitat suitability more generally because they represent a range of habitat needs and are relatively easy to monitor.

Introduction

In South Africa the grassland biome and its associated biota are increasingly becoming threatened due to expansion of agricultural activities, human settlements and associated road infrastructure (Allan et al., 1997; Reyers et al., 2001; Egoh et al., 2011). The growth of the human population in southern Africa is accompanied by increasing demands for water and electricity. These pressures are likely to impact on bird species richness in remote eastern, moist, high-altitude grasslands (e.g., Maphisa et al., 2016). This area is a centre of endemism for both plants and animals (Zunckel, 2003) and has the highest concentration of Important Bird and Biodiversity Areas (Barnes, 1998; Marnewick & Retief, 2015) in southern Africa. These grasslands are currently predominantly used to support livestock farming accompanied by annual burning followed by heavy grazing (Maphisa et al., 2016; Maphisa et al., 2017). However, at the start of the 21st century, the area is increasingly targeted for development of large water schemes (e.g., Davies & Day, 1998) and electricity projects to meet increasing demands for water for human consumption (Maphisa et al., 2016; Maphisa et al., 2017). The network of new roads to access these schemes and associated power grid in turn makes the area attractive for human settlement and intensification of agricultural activities. These developments, if not carefully planned, may result in habitat loss with possible negative impact on biodiversity in general.

Because of socio-political pressure and despite objections from environmental organisations, development in the area may not be completely prevented (Bennett et al., 2017). As a result of these threats facing mountain grassland habitats, there is now an urgent need for biodiversity information to identify and protect habitats for threatened fauna and flora of the area. As the pressure on the remaining natural land increases, these areas also need to be managed more effectively.

Before people started to use these grasslands intensively, herds of roaming wild antelopes are thought to have been responsible for creating a habitat mosaic that created niches for different bird species (Hockey et al., 1988). In recent times, planned man-made fires and grazing by domestic livestock have become important tools that grassland managers can use to manage grasslands for bird diversity. Grazing by mixed livestock and fires of different intensities create a habitat mosaic in grass height and cover which benefits a variety of species across the landscape at different times of the year (Hobbs & Huenneke, 1992; Parr & Chown, 2003; Tews et al., 2004; Vandvik et al., 2005; Fuhlendorf et al., 2006; Evans et al., 2006; Fahrig et al., 2011). Even though grass height and cover appear to be the two key variables determining habitat suitability for grassland bird species (Driscoll et al., 2010), we still have little information on how different species react to grass height and cover throughout the breeding season.

Dynamic site occupancy models were initially developed as an approach to investigate the dynamics of species occurrence and to understand how factors of interest affect the vital rates that determine occurrence (rates of local persistence and colonization) (Mackenzie et al., 2011). Site occupancy models offer opportunities to frame and solve decision problems for conservation that can be viewed in terms of site occupancy (Royle & Kéry, 2007; Martin et al., 2009) and are well suited for addressing management and conservation problems (Martin et al., 2009). Non-detection of a species at a site does not imply that the species is absent unless the detection probability is one (MacKenzie et al., 2003). Occupancy models account for imperfect detection, which is the inability of investigators to detect a species at a site with certainty (Zipkin, DeWan & Royle, 2009). Incorporating detection probabilities into estimates of species richness is important for obtaining unbiased estimates of species numbers, particularly in communities with large numbers of rare or elusive species (Mackenzie et al., 2011; Govindan, Kéry & Swihart, 2012; Oedekoven et al., 2013). Accounting for detectability is particularly important amongst grassland birds because many grassland birds are hard to identify or highly elusive.

In this study, we use repeated detection-non-detection data and state-space dynamic occupancy models, to evaluate how grass height and cover influence habitat use by 12 common bird species in high-altitude grassland in eastern South Africa throughout a summer breeding season. Grass height and cover are proximate factors influencing habitat selection and nest survival amongst grassland birds (Devereux et al., 2008; Whittingham & Devereux, 2008; Cao et al., 2009; Donald et al., 2010; Fisher & Davis, 2010; Klug, Jackrel & With, 2010). Several species of conservation concern occur with conflicting habitat requirements in our study region (Maphisa et al., 2009; Maphisa et al., 2016). In the case where habitat is managed to maximize biodiversity, management actions that enhance habitat for some species may limit habitat for other species.

The goal of this study was to examine the response of 12 common grassland bird species to grass height and cover throughout an austral summer (2011/12) which coincides with high avian species richness and the time when most birds are breeding (Maphisa et al., 2016). We examined how grass height and cover was related to initial occupancy at the beginning of the breeding season, and how these two variables affected changes in habitat use throughout the season.

Materials and Methods

Study area

This study was conducted at Eskom Ingula Pumped Scheme property with a few plots randomly selected from the neighbouring privately owned farms (Fig. 1). Ingula is located c. 23km north-east (28°14′S, 29°35′E) of the village of Van Reenen at altitudes of 1,200 to 1,700 m asl and covers c. 8,000 ha (Maphisa et al., 2016; Maphisa et al., 2017). It straddles the escarpment and two provinces: KwaZulu-Natal and Free State (FS). The average altitude below the escarpment is 1,200 m asl and 1,700 m asl above the escarpment. The FS side is dominated by sweet and sour grassland vegetation type (Mucina & Rutherford, 2006), characterised by the grass Themeda triandra. The area below the escarpment is dominated by Hyparrhenia-Cybompogon grasses and has been modified into fields and alien plantations and therefore is considered of less conservation priority compared to the upper site (Maphisa et al., 2016).

Figure 1 Map of study area where green dots represent centre of each plots.

Map of study area showing location of our 500 m × 500 m random study plots (greendots) include plots on adjacent farms to Ingula property. This map was created using gplot2, lon and lat on the y and x axis both represents longitude and latitude respectively. Google imagery ©2018 TerraMetrics; map data AfriGIS (Pty) Ltd, Google.

Some parts of Ingula property (the area above the escarpment) and surrounding privately owned farms are designated an Important Bird and Biodiversity Area mainly because of high avian endemism (Marnewick et al., 2015). Previous land owners used annual fires to optimize domestic livestock production, the practice that over the years resulted to negative impact on ecosystem and biodiversity (Maphisa et al., 2016). Current Ingula management seeks scientific advice on how to reverse past harmful grassland management practices and make habitat suitable for unique fauna and flora of the area. The surrounding farms are still heavily grazed and annually burned with negative impact on habitats and species. This study was approved by the Ingula Partnership while DHM was an employee of Birdlife South Africa. The Ingula Partnership is made up of Birdlife South Africa, Middelpunt Wetland Trust and Eskom. DHM got further verbal permission from the neighboring farm owners to enter their properties and record birds and vegetation for the duration of this study. Commercial farmers’ cattle were moved out of Ingula since summer of 2005 so that the area could recover from past heavy livestock grazing and annual burning. However, relatively small herds of livestock belonging to the former land owners’ tenants remained on site throughout the duration of our study (Maphisa et al., 2016).

The weather at Ingula is characterised by cold winters with occasional snow and strong directional winds and wet summers dominated by morning mist (Maphisa et al., 2016; Maphisa et al., 2017). Most of the rainfall occurs during the southern hemisphere summer (October to February), sometimes with marked rainfall differences between the upper and the lower parts of the study area. At Ingula, this sharp seasonal contrast in temperatures also affects bird species richness with highest species richness occurring in summer while the winters and spring recorded the lowest species richness (Maphisa et al., 2016). Therefore this study uses summer data for birds and vegetation when species richness is highest.

Vegetation and bird surveys

We laid a grid of 500 m × 500 m on 1:50,000 topographic maps of the entire study area and extended the grid into adjacent neighbouring privately owned farms. We excluded all the plots that were steep or rocky. Slope aspects may compound our effort to associate grass height and cover to plot occupancy as birds tend to prefer certain slope aspects to others (Brambilla et al., 2013). Rocky outcrops also tend to be preferred by other types of birds that are none grassland birds. The topography of our study area is rugged (Maphisa et al., 2016) and bisected by other habitats other than just grassland (Fig. 1). We also excluded plots that were lying adjacent to the wetlands too to avoid associated edge effects (Bazzi et al., 2015). Then we sequentially numbered all the remaining plots and randomly selected 19 of them using the ‘sample()’ function in R (R Development Core Team, 2013). Twelve plots were located within the Ingula property itself and seven plots on neighbouring farms (Fig. 1; plots P03A, P10, P50, P54, P57, P63, P64 were on private land).

On these plots we surveyed birds and vegetation during the austral summer of November 2011 to February 2012, spanning the entire breeding season. Within each plot we walked at constant pace along a continuous straight path inside each plot 150 m from the edge on three sides of the plot till we reached the forth side where we started. Each survey lasted 30 min. We recorded all birds that we saw inside the plot as we walked. Where grazing is excessive or grassland has been recently burned it is easy to see Common Quail Coturnix coturnix on the ground even when not calling (Maphisa et al., 2016). Except for Common Quail, we also recorded species based on call if the call clearly originated from within the plot limits. We assumed that walking rather than point counts would maximize detection of secretive grassland birds such as Yellow-breasted Pipit (Anthus Chloris) or Common Quail. The sighting and recording of birds was done by one person (DHM) with good experience of habitats of birds of this region. Each plot was visited three times each month (November to February). Two of the surveys were undertaken mostly in the mornings, from 07h00–11h00, and sometimes in the afternoons from 15h00–16h00 when weather prevented completion of surveys during the morning (e.g., Maphisa et al., 2016). Weather permitting, we ensured that the repeat surveys were very close to the first survey. During the third survey we also recorded vegetation characteristics in addition to birds and therefore survey mostly took a little longer than 30 min.

We recorded grass height and cover using similar methods as that of (Maphisa et al., 2009; Maphisa et al., 2017). We randomly placed a 30 cm × 30 cm quadrat (divided into nine equal squares) twice every 100 m along the route where we had recorded birds earlier (Maphisa et al., 2017). We recorded grass cover as the number of squares with grass out of the total of nine squares in the frame. Each square that was at least 75% grass was considered grass. We recorded grass height at the four corners of the grid and averaged the four measurements for our analysis. This field protocol enabled the two variables to be measured over a relatively large area (Fig. 1). Weather conditions affect detectability of birds (Zuckerberg et al., 2011; Hovick, Elmore & Fuhlendorf, 2014; Sliwinski et al., 2016). During each survey, we recorded cloud cover (clear, partly cloudy or cloudy) and temperature (cold, cool, warm or hot), together with wind conditions (calm, moderate or strong) (Harms et al., 2017). Because of our small data set, we wanted to reduce these weather covariates into a single variable representing observability. The variables were on an ordinal scale and to us, the most sensible way to reduce them to a single index was by subjectively scoring their effects based on our ability to detect birds (Appendix S1). The purpose of the observability covariate was simply to capture some of the variability in the detection probabilities, not to test any hypotheses about the detection process (Royle, 2006). No survey was carried out when poor visibility would impact the identification of birds.

In this study, we focused on the following 12 species, from the most common to the least common (based on preliminary data analysis): African Pipit Anthus cinnamomeus, Cape Longclaw Macronyx capensis, Wing-snapping Cisticola Cisticola ayresii, Red-capped Lark Calandrella cinerea, Zitting Cisticola Cisticola juncidis, Yellow-breasted Pipit, Common Quail, Long-tailed Widowbird Euplectes progne, African Quailfinch Ortygospiza atricolis, Banded Martin Riparia cincta, Ant-eating Chat Myrmecocichla formicivora and Eastern Long-billed Lark Certhilauda semitorquata. Of these species, the Yellow-breasted-Pipit is considered nationally threatened (Barnes, 2000; Taylor, Peacock & Wanless, 2015). Our study area is home to other threatened species (Maphisa et al., 2016), but those were rare and resulted in low detections. The justification for choosing the 12 species above is that they are all typical grassland species with a diversity of habitat requirements (Maphisa et al., 2017) some of which could share habitat with rare species of management concern.

Model description

We used dynamic multi-species occupancy models to examine habitat use of 12 grassland bird species throughout their breeding season. The main habitat structuring factors we considered were grass height and cover. We were interested in the types of habitat each species occupied and how occupancy changed in relation to habitat from one month to the next over the course of a breeding season while accounting for imperfect detection. The basic idea is that (1) non-detection can be distinguished from absence through repeated sampling and (2) species-specific estimates of occurrence can be improved using collective data on all species observed during sampling (Zipkin et al., 2010). The dynamic model describes occupancy as a state process based on: (1) persistence: the probability of an occupied site continuing to be occupied from one month to the next, and (2) colonisation: the probability of an unoccupied site becoming colonised (Popescu et al., 2012).

We developed a multi-species hierarchical model (Appendix S2) using a state-space formulation (Royle & Kéry, 2007). Our model assumes that site-specific occupancy (i.e., ‘true’ detection/nondetection) for species i = 1, 2, …, N at site j = 1,2.., J, is denoted zi,j, where zi,j = 1 if species i occurs at site j and otherwise zi,j = 0 (Zipkin, DeWan & Royle, 2009). Therefore, the occupancy status of species i at site j during repeat k visit month is represented by zi,j, where occupancy status of of each site is assumed to be constant across the three monthly repeat surveys. The occurrence status of each species is the stochastic binary outcome governed by the occupancy probability (Ψ) of species i at site j which is assumed to be the outcome of Bernoulli random variables such that zi,j ∼Bernoulli (Ψi,j), the probability that species i is present at site j is represented by Ψi,j (Dorazio et al., 2006).

Similarly, we assumed that a species can only be detected at a site if it actually occurs there, i.e., there are no false positives (e.g., Dorazio & Royle, 2005; Dorazio et al., 2006). Therefore, detection of species i at site j on visit k is represented by detection probability pi,j,k with occupancy state: xi,j,k∼Bernoullipi,j,k×zi,j,k.

(Dorazio et al., 2006; Russell et al., 2009; Zipkin, DeWan & Royle, 2009).

We were interested in the seasonal changes in the bird communities and therefore we used a dynamic extension of the model above, allowing the occupancy status to change from one month to the next (e.g., Iknayan et al., 2014). We modelled occupancy during the first month (November, t = 1) as above, zi,j,t∼BernoulliΨi,j,fort=1.

Occupancy during the subsequent months depended on occupancy during the preceding month: zi,j,t |zi,j,t−1,φi,j,t,γi,j,t∼Bernoulliφi,j,t×Zi,j,t−1+γi,j,t×1−Zi,j,t−1,fort>1,

where the colonisation probability (γ) is the probability of an unoccupied site becoming occupied and the persistence probability (φ) is the probability of an occupied site remaining occupied (Royle & Kéry, 2007). The occupancy probabilities during December, January and February (t = 2, 3, and 4) were calculated as derived parameters.

Initial occupancy, colonisation and persistence were constrained to be linear functions of the covariates grass height (avh) and grass cover (cover) on the logit scale: LogitΨi,j=β0j+β1j×avhi,j,t+β2j×coveri,j,tfort=1

Logitγi,j,t=ν0j+ν1j×avhi,j,t+ν2j×coveri,j,tfort>1

Logitφi,j,t=μ0j+μ1j×avhi,j,t+μ2j×coveri,j,tfort>1,

where the β, ν and µ are species-specific coefficients. Each of these nine coefficients was modelled as a separate random effect, i.e., ηi ∼N(η_bar, ση) where η_bar is the mean and ση the standard deviation of the species-specific coefficients and η = {β0,1,2, ν0,1,2, µ0,1,2}.

As with occupancy (Kéry et al., 2009; Russell et al., 2009), we incorporated factors that influence detection of each species during each site visit. We modelled the detection probability (p) as a function of conditions measured by the continuous covariates obs (appendix), and as a function of the day of the survey (day: the number of days since the start of the season). α0, α1 and α2 as coefficients: Logitpi,j,k,t=α0+α1∗obsi,j,k,t+α2∗dayi,j,k,t

(Zipkin, DeWan & Royle, 2009; Zipkin et al., 2010).

Each of the three coefficients was modelled as a separate random effect, i.e., αj ∼N(α_bar, σα) where α_bar is the mean and σα the standard deviation of the species-specific coefficients.

Each covariate was centred and scaled before analysis (Nichols & Boulinier, 1998; Van den Berg et al., 2006; Jones et al., 2012; Pollock et al., 2014). We then calculated the number of species (Dorazio et al., 2006; Zipkin, DeWan & Royle, 2009), out of the 12 studied species, that are present at a site in a given month (local species richness, ri,t = Σjzi,j,t) and the number of plots each species occupied in a given month (oj,t = Σizi,j,t) as derived parameters.

Model fitting and analysis

We estimated the parameters using a Bayesian analysis of the model with vague priors (Royle & Kéry, 2007; Russell et al., 2009; Zipkin, DeWan & Royle, 2009; Banks-Leite et al., 2014) for all parameters. We used a flat normal distribution N(0,100) for the means of the coefficients and Inverse Gamma (0.01,0.01) for the variances of the random effects. We tested the sensitivity to the choice of priors for the latter by also using U(0,15) as priors for the standard deviations (Zipkin, DeWan & Royle, 2009) (Appendix S2).

We carried out the analysis in JAGS (Plummer, 2003) called via package rjags (Plummer, 2014) from R (R Development Core Team, 2013). The MCMC procedure requires an initial burn-in period for the chains to converge to a stationary process, after which the subsequent values can be used to calculate medians and credible intervals associated with the parameters of interest (Sauer et al., 2013). We assessed convergence by verifying that the Gelman–Rubin statistic was below 1.1 (Brooks & Gelman, 1998; Gelman & Shirley, 2011) and visual inspection of the chains (Jones et al., 2012). We ran three chains of length 60,000 each; with a burn-in of 30,000 and thinned the remaining results by taking each 20th value from the chains. With these settings, the model converged for all parameters.

Results

The detection probabilities varied among species and increased with increasing magnitude of our observability covariate for most species, indicating that the variable captured some of the variability in detection probabilities (Fig. 2). For most species, the detection probability also increased over the course of the season (Fig. 2).

Figure 2 Effect of the two covariates on detection of the 12 birds.

Effect of average estimated prevailing weather conditions at the time of survey (observability) and date of survey on detection of each of the 12 species throughout the four austral summer months. Ap, African Pipit; CLc, Cape Longclaw; WsC, Wing-snapping Cisticola; RcL, Redcapped Lark; ZC, Zitting Cisticola; YbP, Yellow-breasted Pipit; CQ, Common Quail; LtW, Longtailed Widowbird; AQf, African Quailfinch; BMtn, Banded Martin; AEC, Ant-eating Chat; ELL, Eastern Longbilled Lark.

Species occupancy phenology over the four months

Plot occupancy was variable among the 12 species across the four months, with overall high initial occupancy followed by a gradual decline in the number of occupied plots for a majority of the 12 species as the season progressed (Fig. 3). The Wing-snapping Cisticola was recorded in almost every plot throughout the four months and occupancy for this species was estimated to be 1. Four other species, Cape Longclaw, African Pipit, Zitting Cisticola and Banded Martin, exhibited high plot occupancy too throughout the four months. Three other species were common early during the season but showed a rapid decline to a low number of occupied plots by the fourth month (number of occupied plots in November minus number of occupied plots in February [95% credible interval]: Red-capped Lark 14 [12, 15]; Common Quail 11 [5, 16], Yellow-breasted Pipit 9 [5, 11]). Long-tailed Widowbird and Eastern Long-billed Lark occupied the fewest number of plots throughout the season, with Eastern Long-billed Lark showing a rapid decline between the third and fourth months (Fig. 3, number of occupied plots in January minus number of occupied plots in February 7 [0, 10]).

Figure 3 Number of plots occupied by each species out of 19 plots.

Estimated number of plots occupied (out of 19 plots) for each of the 12 bird species across the four summer months.

Effects of habitat on occupancy, persistence and colonisation

On average across species, initial occupancy increased with increasing grass height and decreased with increasing grass cover (heavy black lines in Fig. 4). The species therefore tended to prefer relatively open plots with high grass at the beginning of the breeding season. Over the course of the season, they tended to have a higher persistence on patches with short grass and little cover, and were more likely to colonize such plots. However, there was considerable variation among species in their responses to grass height and cover.

Figure 4 Effects of grass height and grass cover on occupancy.

Hierarchical dynamic plot occupancy of the 12 most common species, showing influence of grass height and cover on persistence and colonization of each species during the austral summer survey during 2011/12 at Ingula. The thick place line represent average mean. The species abbreviations represent the following birds; AP, African Pipit; CLc, Cape Longclaw; WsC, Wingsnapping Cisticola; RcL, Redcapped Lark; Zitting Cisticola; YbP, Yellow-breasted Pipit; CQ, Common Quail; LtW, Longtailed Widowbird; AQf, African Quailfinch; BMtn, Banded Martin; AEC, Ant-eating Chat; ELL, Eastern Longbilled Lark.

One group of species (Cape Longclaw, Wing-snapping Cisticola, Zitting Cisticola and Banded Martin) occurred on most plots throughout the season and consequently showed little sensitivity to grass height and cover. Long-tailed Widowbird and Eastern Long-billed Lark were two of the species with the strongest preference for plots with open and high grass, and they tended to persist better in such plots. Among the two species that showed biggest decline in occupancy through the season, Red-Capped Lark was more likely to persist on plots with short and open grass, whereas Common Quail had relatively higher persistence on plots with short and dense grass. Of the two species that tended to increase occupancy over the season, African Quailfinch was more likely to colonise plots with short dense grass and Long-tailed Widowbird was more likely to colonise plots with short and open grass.

Species richness: comparing Ingula with neighbouring private farms

Species richness did not vary much over the four summer months and was similar on Eskom’s property compared to neighbouring private farms in November (difference [95% credible interval] in species richness: 0.5 [−0.3, 1.3]) and December (−0.3 [−0.9, 0.3]) but slightly higher on private land in January (1.2 [0.5, 1.8]) and February (0.8 [0.1, 1.5], Fig. 5).

Figure 5 Response of each of the 12 species to grass height and grass cover.

Bird species richness comparing Ingula farms that experienced relatively little grazing compared to neighbouring farms which were intensively grazed with cattle during the four summer months. Almost all plots both inside Ingula property and on adjacent farms were burned prior to summer but sometimes late into the summer breeding season.

Discussion

The primary aim of our study was to investigate how grass height or grass cover or both potentially affect plot occupancy by the 12 common grassland birds of our study. Our results are consistent with earlier studies demonstrating an avifaunal shift in response to change in vegetation height (Martin & Possingham, 2005; Tichit et al., 2007; García et al., 2007) as season progresses. Both grass height and cover are proximate factors which, when managed with fire and grazing, create habitat suitability for birds of contrasting habitat needs (Fuhlendorf, Engle & Moreira, 2004; Maphisa et al., 2016; Maphisa et al., 2017). Our results are consistent with the results of previous studies showing that a habitat mosaic in grass height and cover can allow species differing in habitat needs to occur in the same area (Fuhlendorf, Engle & Moreira, 2004; Coppedge et al., 2008; Hovick, Elmore & Fuhlendorf, 2014). We therefore suggest that the management of this grasslands use fire and grazing pro-actively to increase habitat suitability that would benefit species of contrasting habitat requirements (Fuhlendorf et al., 2009). Such management should also benefit rare species for which we did not have enough data to include in this analysis.

We examined how environmental factors and the time of day in our austral summer season affect detection of selected typical grassland birds. We found the detection of the following species showing pronounced decline with season (African Pipit, Long-tailed Widowbird, African Qualifinch and Ant-eating Chat) in an environment where the breeding season is short and birds undertake local altitudinal migration (Berruti, Harrison & Navarro, 1994). Some species could be harder to detect later in the season because they are less active when breeding or as grass height grows taller compared to early in the season (Furnas & McGrann, 2018; Grinde, Niemi & Etterson, 2018) in an area which is mostly annually burned before summer (Maphisa et al., 2016; Maphisa et al., 2017). Overall, our results on effect of weather and date of survey in a season seem to largely support each other (e.g., Fig. 2).

Amongst these species, the threatened Yellow-breasted Pipit and Red-capped Lark could serve as indicators of habitat suitability for other species under managed grazing and burning. For example, although Yellow-breasted Pipit indicate a high initial plot occupancy (Fig. 3) and high plot occupancy across average grass heights (Fig. 4), this species reacts negatively to increase in grass cover (Fig. 4). This indicate a need for controlled grazing post burning to open up patches for feeding and nesting. Initially, Red-capped Lark indicates a high plot occupancy (Figs. 3 & 4) but low persistence with increasing grass height and grass cover. Therefore, high plot occupancy of this species during mid-season when birds are breeding, would indicate that average grass height is too short and open, indicating overgrazing. Overgrazing leads to low nest survival amongst grassland birds and it must be avoided in this grasslands (Maphisa et al., 2009).

The increasing demand for land for development necessitates more effective management of the remaining ecosystems and biodiversity (Zipkin, DeWan & Royle, 2009; Drum et al., 2015). Detection-nondetection data and multi-species occupancy models (MacKenzie et al., 2003; Popescu et al., 2012) provide a cost effective way of studying the response of a collection of species for management of the habitat (Sauer et al., 2013).

We modelled occupancy, persistence and colonisation as logit-linear functions of grass height and cover from one month to the next for four months (multi-seasons). This was a simple approach. An alternative approach would have been to consider models with quadratic terms (Zipkin, DeWan & Royle, 2009; Zipkin et al., 2010; Ruiz-Gutiérrez, Zipkin & Dhondt, 2010) to examine species-specific optima in grass height and cover. However, we did not do this due to small sample size (12 plots within Ingula compared to 7 on private land), which was a result of the study site rugged topography and the fact that the surveys were undertaken by one person. Even with five plots short, the mean species richness comparing the two differently managed sites is about the same. This is because each farmer on the neighbouring farms applies fire and stocking densities differently and therefore provides heterogeneity in grass height and cover which would benefit different species.

Hierarchical multi-species occupancy models that require detection-nondetection data (e.g., Goijman et al., 2015) have advantage compared to other methods (e.g., Maphisa et al., 2017) for monitoring purposes because only presence/absence of species is recorded. Repeated surveys provide important information on the observation process (Kery, Guillera-Arroita & Lahoz-Monfort, 2013; Sauer et al., 2013). The difficult topography of our study area plus inclement weather made it difficult to have large sample size. However, despite the small number of plots surveyed, our findings have provided management of these grasslands with important vegetation attributes that need to be manipulated with fire and grazing to make habitat suitable for birds. Recent improvements in hierarchical multi-species dynamic occupancy models (e.g., Bailey, Mackenzie & Nichols, 2014) makes it possible during monitoring to predict areas that may not be sampled in some years due to logistic factors such as inclement weather (e.g., Iknayan et al., 2014).

Supplemental Information

Appendix S1 The way observability covariates were calculated

Scoring of prevailing weather conditions (observability) each weighted according to how DHM based on knowledge of the area perceived a weather variable to nfluence detection of birds. Conditions were optimal with a clear sky (score 1 = 100), cool emperatures (score 2 = 100) and calm wind conditions (score 3 = 100). For other weather conditions, observability was reduced and we chose the scores according to our subjective.

Click here for additional data file.

Appendix S2 R and Bugs code used to fit the model

Click here for additional data file.

Supplemental Information 1 Detection-nondetection xls file of birds plot data, only summer data was used

Click here for additional data file.

Supplemental Information 2 Vegetation plot data representing three consecutive bird surveys

Click here for additional data file.

We are greatly indebted to Sakhile Wiseman Mthalane who assisted (DHM) during fieldwork. Also, we would like to thank farmers whose property boarders Ingula and allowed us unrestricted access to their land during our field work surveys. We thank Prof Les Underhill for providing the critique that helped us to make improvements on our original manuscript. We are greatly indebted to Malcolm Drummond for proofreading the original manuscript and providing comments.

Additional Information and Declarations

Competing Interests

Author Contributions

Field Study Permissions

Data Availability

Both David H. Maphisa and Hanneline Smit-Robinson were employees of BirdLife South Africa at the time of this study. The authors declare there are no competing interests.

David H Maphisa conceived and designed the experiments, performed the experiments, analyzed the data, contributed reagents/materials/analysis tools, prepared figures and/or tables, authored or reviewed drafts of the paper, approved the final draft.

Hanneline Smit-Robinson authored or reviewed drafts of the paper, approved the final draft.

Res Altwegg analyzed the data, contributed reagents/materials/analysis tools, prepared figures and/or tables, authored or reviewed drafts of the paper, approved the final draft.

The following information was supplied relating to field study approvals (i.e., approving body and any reference numbers):

This study was approved by Ingula Partnership while DHM was employee of Birdlife South Africa. Ingula Partnership was made up of Birdlife South Africa, Middelpunt Wetland Trust and Eskom. DHM got further verbal permission from the neighboring farms to enter their properties for the duration of this study to record birds and vegetation

The following information was supplied regarding data availability:

Dryad: DOI 10.5061/dryad.pn3623b.

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
