# Peer review of "Dynamic multi-species occupancy models reveal individualistic habitat preferences in a high-altitude grassland bird community"

_PeerJ, doi:10.7717/peerj.6276_

## Round 0.1 · original submission · Minor Revisions

This is quite a well-written and clear paper. However, the reviewers identified some areas for improvement. The authors should revise the introduction in an effort to identify the knowledge gap being investigated. I also agree with reviewer 1 that measures of goodness of fit of the model should be provided. Reviewer 2 asks that the take-home message be identified and strengthened.

Reviewer 1 ·

Basic reporting

I found this manuscript describing the results of a multi-species dynamic occupancy model of highland grassland birds in southern Africa to be very nicely crafted and an enjoyable read. I provide below a few general comments but then spend most of my time in this review providing suggestions for clarifying the text.
Starting at line 67 and going through line 87, methods of analysis are presented before the questions being addressed are provided to the reader. Are the questions motivated by the methodological approach, or are the questions dictating the approach to answering them? If the latter, this arrangement in the Introduction should be altered to move the focus to the questions (i.e., the role of grass height and cover on species occurrence). One simple solution is moving the paragraph in lines 88-96 before the paragraphs starting at line 67.
The authors’ approach allows for estimation of detection probability, but I can find no description of those results. The effect of obs is not described, nor are species-specific detection probabilities provided. I think it would be instructive for p to be described, perhaps akin to Figure 2 and 3.
As it currently stands, the reader has no idea whether the model fits the data well. There are no goodness of fit assessments. An omnibus goodness‐of‐fit test using a chi‐square discrepancy measure on unique detection histories is available for occupancy models (MacKenzie and Bailey, Journal of Agricultural, Biological, and Environmental Statistics, 9, 2004, 300; the MacKenzie–Bailey test). Broms et al. suggest WAIC, CPO, and other methods (Broms, Kristin & B. Hooten, Mevin & Fitzpatrick, Ryan. (2016). Model selection and assessment for multi-species occupancy models. Ecology. 97. 10.1890/15-1471.1.).

Line-by-line comments
The third (3) affiliation, i.e., Ingula Partnership Project etc., seems to be missing from the article title page, but it is available in the separate abstract page preceding the manuscript.
Line 34. ‘Grazing’ to ‘grazing’.
Line 47. Is the comma after the parenthetical necessary?
Line 87. ‘set management objectives’? Is ‘set’ necessary?
Line 98. Suggestion: ‘maintain long-term avian diversity.’
Line 128. ‘with a few plots’.
Lines 134-136. Unclear. Suggested rewrite: ‘This area, previously used to support commercial livestock in summer, falls partly…’
Line 163. Is ‘lie’ necessary?
Line 169. The citational support here is incorrectly formatted.
Line 170. Only birds walking in and out of the plot were recorded? What kind of birds are these? Quail-like animals?
Line 173. ‘Each birds’? Each species? Each individual bird?
Line 174. Assuming bird should be singular, the verb tense should be ‘was’.
Line 183. ‘completion of surveys’
Line 184. Close in time or space?
Line 185. ‘a similar method’ or ‘similar methods’
Line 186. ‘This method’. No naked this.
Line 186. Throwing? Placing?
Line 216. Space before parenthetical.
Lines 221-222. Italicize Coturnix coturnix.
Lines 230-233. There seems to be a verb missing. Perhaps after ‘site i’ should be ‘site i that is assumed to be’?
Line 237. Period after Bernoulli distribution should be removed.
Line 247. i,j,t should be subscripted for γ.
Lines 248 and 249. Perhaps ‘to become’ and ‘to remain’ should be changed to ‘becoming’ and ‘remaining’?
Lines 260-266. There is no connection, as currently written, for p back to Θ. I think you require a line like 245, but written for Θ rather than z, and p rather than Ψ.
Line 260. For clarification, it would be best to insert ‘month-specific’ prior to ‘detection probability’, given that p is indexed by t.
Lines 294 onward. This section is awkward in that it bounces among the different population parameters. My initial thought to make it more sensible would be to compartmentalize the parameters into their own paragraph. For instance, the first paragraph of this section begins with occupancy, moves to a general description among the three parameters, spends a fair bit of time muddling persistence and colonization, and then circles back in the next paragraph back with occupancy. Then, halfway through that paragraph, it veers toward persistence and colonization. It’s hard to keep track the way it is written. In retrospect, maybe what it requires is a sentence ahead of the second paragraph that moves from the generalization of results to the species-specific findings, something along the lines of… “Species differed in their response to grass height and cover.”
To reiterate, because I’m having so much trouble here… two sentences in the first paragraph seem to conflict. Starting at line 296, “Across the 12 species, … colonization decreased with increasing grass height and cover…”, Starting at line 301, “Overall, colonization declined with increasing grass height and cover for a majority of the 12 species, with [3 species] being exceptions…” The first sentence suggests there are no exceptions, the section sentence says the exact same thing but identifies 3 exceptions.
Line 295. Insert ‘the’ or ‘a’ prior to ‘majority’.
Lines 306-307. ‘little’ should be moved before ‘affected’.
Line 334. I’d replace ‘monitoring’ with ‘assessing’, as the models interpret results of the monitoring.
Line 346. Cover the naked this by inserting ‘result’ after ‘this’.
Line 347. Replace ‘nest’ with ‘next’.
Line 347. Insert ‘the’ or ‘a’ prior to ‘majority’.
Line 348. Reconcile verb tense. ‘this result supports’ or ‘these results support’.
Line 353. Is ‘and’ necessary after ‘burning and grazing’? Replace with comma? And then insert a comma after ‘unburned’.
Line 369. Replace ‘resulted to’ with ‘resulted in’.
Line 395. Replace ‘first’ with ‘most’.
Line 395. Cover the naked this by inserting ‘result’ after ‘this’.
Line 402. There seems to be a formatting problem here. An errant space occurs before the period.
Line 346. Cover the naked this by inserting ‘approach’ after ‘this’, and the subsequent ‘approach’ with ‘one’.
Line 346. Cover the naked this by inserting ‘expansion’ after ‘this’.
Line 436. This sentence is muddled. Models are not a survey method, and ‘purposed’ does not work here.
Line 440. Rectify verb tense. ‘Improvements… make’ or ‘Improvement… makes’.
Figure 3. I have a hard time differentiating these colors. There is a thick black line that, I surmise, is supposed to be the independent mean of the species estimates, but this is described nowhere.
Figure legend 4 misspells ‘comparing’.

Experimental design

Nothing further to add here.

Validity of the findings

Here, beyond the comments already given, I wish to reemphasize the need for model assessment. Further, the authors should report their results pertaining to detection probability.

Additional comments

No additional comments.

Reviewer 2 ·

Basic reporting

Clear and unambiguous, professional English used throughout: Use of English could be improved in some places, but is generally good. Specific comments in attached PDF.

Literature references, sufficient field background / context provided: Introduction needs substantially more background / context for scientific questions. Use of primary literature could be improved in the discussion. Suggestion for improvement included in attached PDF.

Professional article structure, figs, tables. Raw data shared: Some figures could be improved, particularly Figure 3. Specific suggestions included in attached PDF. Thank you for sharing raw data. Code used to format raw data for analysis in JAGS would be helpful.

Self-contained with relevant results to hypotheses: I had a hard time finding the take-home message. The discussion didn't relate well to the objectives defined in the last paragraph of the introduction. i.e., I didn't have a good sense of the response of the 12 common species to grass height and cover, or how I could use that information to inform grassland management. Specific comments in attached PDF.

Experimental design

Original primary research with Aims and Scope of the journal: This appears to be original primary research.

Research question well defines, relevant & meaningful. It is stated how research fill an identified knowledge gap: It is not clear to me how the research fills an identified knowledge gap. No gap is identified. A more thorough literature review in the introduction is necessary.

Rigorous investigation performed to a high technical & ethical standard: Research appears to conform to prevailing ethical standards. No animals are handled, and the author indicates he / she had permission to survey on private lands. Uses state of the art dynamic occupancy models, though there is no evaluation of potential competing models. See attached PDF for details.

Methods described with sufficient detail & information to replicate: Substantially more information needed in methods section to replicate study. Details are in attached PDF.

Validity of the findings

Data is robust, statistically sound, & controlled: Sample size is small (~ 20 sites). The author probably needs to use more informative priors if they want to derive meaningful inference from their data. I need more details on study design to determine if statistically sound. Data are not controlled in terms of burning and grazing, which seems to be an area where the authors wish to make inference. This can be okay from a design perspective, since this is an observational study, but authors do not include fire / grazing covariates in their models.

Conclusions are well stated, linked to original research question & limited to supporting results: Not always. For example, the authors make recommendations regarding burning and grazing, but do not include burning and grazing in their models. Details in attached PDF.

Speculation is welcome, but should be identified as such: Some speculation in discussion, not substantiated by prior literature or identified as such. For example, the authors speculate when describing causes for decline in occupancy on line 387, without reference to the primary literature. Details in attached PDF.

Additional comments

See attached PDF

Annotated reviews are not available for download in order to protect the identity of reviewers who chose to remain anonymous.

---

## Round 0.2 · accepted · Accept

Thank you for your thorough revision of this paper. I believe it is acceptable for publication.

#